# Use of ECMO for Cardiogenic Shock in Pediatric Population

**DOI:** 10.3390/jcm10081573

**Published:** 2021-04-08

**Authors:** Priya Bhaskar, Samuel Davila, Aparna Hoskote, Ravi Thiagarajan

**Affiliations:** 1Division of Pediatric Critical Care, UT Southwestern Medical Center, Children’s Medical Center, Dallas, TX 75235, USA; priya.bhaskar@utsouthwestern.edu (P.B.); samuel.davila@utsouthwetsern.edu (S.D.); 2Cardiac Intensive Care Unit, Great Ormond Street Hospital for Children NHS Foundation Trust, Great Ormond Street, London WC1N 3JH, UK; aparna.hoskote@gosh.nhs.uk; 3Department of Cardiology, Division of Cardiovascular Critical Care, Boston Children’s Hospital, Harvard Medical School, Boston, MA 02115, USA

**Keywords:** ECMO, cardiogenic shock, congenital

## Abstract

In children with severe advanced heart failure where medical management has failed, mechanical circulatory support in the form of veno-arterial extracorporeal membrane oxygenation (VA ECMO) or ventricular assist device represents life-sustaining therapy. This review provides an overview of VA ECMO used for cardiovascular support including medical and surgical heart disease. Indications, contraindications, and outcomes of VA ECMO in the pediatric population are discussed.VA ECMO provides biventricular and respiratory support and can be deployed in rapid fashion to rescue patient with failing physiology. There have been advances in conduct and technologic aspects of VA ECMO, but survival outcomes have not improved. Stringent selection and optimal timing of deployment are critical to improve mortality and morbidity of the patients supported with VA ECMO.

## 1. Introduction

Mechanical circulatory support is an essential tool in the care of children with severe refractory cardiac failure, acute or chronic, when conventional medical management has failed. The two primary forms of mechanical circulatory support are veno-arterial extracorporeal membrane oxygenation (VA ECMO) and ventricular assist devices (VAD). The two technologies have unique advantages and disadvantages and could be considered as complementary devices and, at times, used in sequence as the clinical scenario dictates [1]. In this review, we provide a current overview of neonatal and pediatric ECMO in the setting of cardiogenic shock, with focus on indications and contraindications, use in special situations, and outcomes.

## 2. Historical Perspective

Based on John Gibbons’ pioneering work on the development of heart-lung machine for cardiac surgery, Bartlett and his colleagues developed and utilized ECMO for an extended period to support critically ill neonates with respiratory failure in the 1970s [1]. This was followed by support for circulatory failure after surgeries for congenital heart disease [2,3]. As the technological aspects of ECMO circuitry have become more sophisticated, the survival outcomes initially improved, the indications expanded, and utilization increased. The summary provided in July 2020 by the Extracorporeal Life Support Organization registry reported 21,829 cases of cardiac failure in neonates and children, accounting for 30% of all cases and a steady increase in the annual ECMO runs over the years [4] (Figure 1). The survival to hospital discharge is reported at 50% and has remained the same through decades [4,5]. Our review highlights some of the selected publications on VA ECMO use and management.

The primary indications for cardiac ECMO can be divided into two main categories: medical and surgical. Table 1 describes theses indications, outcomes, and other considerations.

## 3. Principles in ECMO Management

The physiologic state of the failing heart regardless of the etiology is of bilaterally increased preload and low cardiac output due to impaired contractility. The primary emphasis of ECMO support is to restore end-organ perfusion. Every effort must be taken to identify the cause of decompensation and reason for the ECMO support including identification of residual lesion in setting of post cardiotomy ECMO. Predictors for the use of ECMO for medical indications include elevated markers of cardiac injury (troponin), elevated markers of congestion (BNP), and higher levels of inotropic support [6,10,15,18,22,23]. Predictors for the use of ECMO for surgical indications include lower gestational age, lower body weight, higher complexity of congenital heart disease, and longer cardiopulmonary bypass times [24,25,26,27,28,29,30,31,32].

In VA ECMO, the right heart is decompressed via the venous cannula while the left heart is at risk for congestion and elevation of left atrial or left ventricular end-diastolic pressure if there is minimal or no ejection across the aortic valve. Left atrial hypertension results in pulmonary congestion and can lead to significant pulmonary dysfunction. In order to protect the lungs, the left heart is decompressed in a number of ways that are addressed in the special considerations section. When cardiac recovery is delayed, determination of heart transplant or VAD candidacy is indicated.

VA ECMO can occur through peripheral or central cannulation. Central cannulation is a strategy employed when there has been recent sternotomy, single ventricle lesions secondary to cavo-pulmonary connections, or sepsis where high flow rates may be necessary [33]. Peripheral access is used in the setting of emergent cannulation, or when adequate flows for patient rescue can be achieved with limitations of cannula size.

Anticoagulation is necessary in the use of VA ECMO to prevent thromboses both in the circuit and in the patient. In the setting of post-cardiotomy, where the risk of bleeding is inherently high, the management of anticoagulation is challenging and may be avoided for periods of time. Monitoring of anticoagulation has evolved over time with the development of novel techniques such as thromboelastography. New medications for anticoagulation have become available, yet the relative risks and benefits in pediatric VA-ECMO are unknown.

The contra-indications to VA ECMO include severe coagulopathy, irreversible neurological deficit, extreme prematurity, extreme low birth weight, major genetic abnormalities, co-morbidities, or irreversible primary disease state. In a patient who is unlikely to have cardiac recovery, the use of VA ECMO without an exit strategy such as long-term bridging with VAD for heart transplant, would not be appropriate.

## 4. Medical Indications for VA ECMO

While VA ECMO for post-surgical cardiogenic shock has represented much of the use of extracorporeal life support (ECLS), there are multiple indications for the use of VA ECMO in patients with cardiac failure [5]. Common indications for the use of VA ECMO for cardiovascular support in pediatrics include myocarditis, cardiomyopathy, intractable arrhythmias, and sepsis. Novel indications include diseases associated with SARS-CoV-2. With the advent of VADs and other catheter-based interventions, the conduct of ECMO has changed in relationship to the management of cardiomyopathy and myocarditis. We will review each of these indications and associated outcomes (see Table 1 for summary).

### 4.1. Myocarditis

Myocarditis is an infective process involving the myocardium leading to systolic dysfunction and/or dysrhythmias. This is usually a self-limited process, although the disease may become so severe as to require ECMO. A significant portion of these patients will “burn out” and develop a persistent cardiomyopathy and may need transplantation or mechanical circulatory support. Several investigators have described ways to predict which children may have hemodynamic compromise using clinical data (tachycardia, tachypnea, incidence of dysrhythmias) [7,34] or biomarkers such as troponin and creatinine kinase [35,36]. Rates of ECMO use in myocarditis are estimated at 36% in a prospective long-term registry in pediatric myocarditis in Germany [8]. This may be an over-estimate as in this study the diagnostic criteria were stringent and approximately one-third of the cohort had acute fulminant myocarditis. Fulminant myocarditis represents a subset of patients with myocarditis, and is an acute, severe inflammatory process of the myocardium that leads to myocardial necrosis, cardiogenic shock and subsequent end organ failure [37]. This is often heralded by the development of arrhythmia or ST segment changes in a non-vascular distribution and can lead to acute cardiovascular collapse. In these patients, early recognition of disease and institution of ECMO allow for preservation of organ function and avoidance of cardiac arrest.

The clinical outcomes of children who require ECMO for fulminant myocarditis are promising [34,38]. In a recent meta-analysis, Xiong et al. described 172 children with fulminant myocarditis who required ECMO [39]. Survival to hospital discharge was between 53.8 and 83.3% with cumulative rate at 58.7%. In a single center study over 12 years published by Teele et al., 20 children with acute fulminant myocarditis were described with 10 patients requiring VA ECMO support [34]. Those patients who required ECMO had worse markers of end-organ function, elevated lactate, and were more likely to have had CPR. There were three deaths in this series, all in the group that required VA ECMO. Of the survivors, in one patient VA ECMO was used as a bridge to transplant, while in another it was used as a bridge to VAD and ultimately transplant. Investigators for the Pediatric Interagency Registry for Mechanical Circulatory Support (Pedimacs) describe 31 children (11%) who had myocarditis and received a VAD over a 5-year period [40]. In the German registry of myocarditis, overall mortality was 4.6% with a mortality rate of 21.4% in those who required VADs. A novel concept in circulatory support is the use of Impella, a percutaneous mechanical circulatory support device, as sole rescue device, bridge to ECMO, or adjunct for LV decompression in patients with cardiogenic shock including myocarditis [41,42]. There are limitations to Impella: the smallest patient supported in the pediatric clinical trial was 15 kg, it is only intended for short term support, less than 6 h although some centers will go beyond that, and its use is often associated with hemolysis [36]. Given high rates of survival in patients with myocarditis supported with VA ECMO and the increasing availability of VADs, mechanical circulatory support of this condition remains a reasonable therapy.

### 4.2. Cardiomyopathy

VA ECMO has been used in the setting of cardiomyopathies with the best description by Almond et al. [5]. The ELSO registry and Organ Procurement and transplant network database were reviewed between 1994 and 2009; 218 children with cardiomyopathies were supported with VA ECMO and ultimately listed for transplantation. In this cohort, 24% died on VA ECMO, 29% died on the waiting list and 39% died prior to hospital discharge after transplant. While worsening end organ dysfunction due to cardiomyopathy remains an indication for VA ECMO, it is more frequently used in acute resuscitation, ECPR, or as a bridge to more durable mechanical circulatory support in that cohort. With the advent of the Berlin EXCOR device in the early 2000s and the increasing use of other VADs, the use of ECMO for rescue in this population is likely to decrease. The Berlin EXCOR device was the first FDA approved VAD in children in the United States for support of both the right and left ventricle. The Berlin EXCOR device utilizes a pneumatic pump that is external to the body where some newer devices use centrifugal pumps that are housed within the body. The outcomes of patients who require ECMO either as a bridge to transplantation or VAD have poorer prognoses compared to those who do not in terms of mortality, rates of re-operation, dialysis, infection, and stroke [5,43,44]. In many institutions, a patient who presents with cardiomyopathy who is refractory to medical therapy will go to primary VAD rather than ECMO.

### 4.3. Intractable Arrhythmia

In pediatrics, there is only case report and case series data [9] related to the use of ECMO in the setting of tachycardia induced cardiomyopathy or for recalcitrant primary arrhythmias, as these are reversible conditions. Although primary arrhythmia represents some of the patients who require VA ECMO, arrhythmia secondary to myocarditis or another cardiomyopathy are more frequent and discussed above.

### 4.4. Sepsis

Sepsis represents complicated pathophysiology that includes myocardial dysfunction as well as vasoplegia. Supporting children with this physiology is challenging as the flow requirements may be high. The flow provided by ECMO is limited by patient size, catheter size, hemolysis, and other factors yet ECMO remains as a modality in the treatment of sepsis in recent guidelines [11]. Conventionally, it is thought that the flow requirements for infants and smaller children are more easily accommodated as compared to older children.

In an elegant study using the National Inpatient Sample, the use of VA-ECMO in children with sepsis was associated with considerable resource utilization, but had 59% survival to discharge [10]. A novel approach to the use of ECMO for sepsis is central cannulation such that higher flow rates may be obtained. MacLaren et al. report using this strategy in 23 children with refractory septic shock with excellent outcomes—survival to decannulation in 78% and survival to hospital discharge in 74% [45]. Given this data, more centers consider central ECMO when peripheral ECMO provides inadequate support in the management of refractory septic shock.

### 4.5. SARS-CoV-2

SARS-CoV-2 has been demonstrated in the myocardium of an adult and pediatric patient with myocarditis [12,46,47]. While there is experience in the use of VA and VV-ECMO in the support of respiratory disease or the primary infection in adults, it is extremely infrequent in children. Recently, Di Nardo et al. described the use of ECMO for SARS-CoV-2 in children through a collaborative EuroELSO prospective survey [48]. Seven children were supported with VA ECMO, and one-third had significant co-morbidities. Four were supported for primary respiratory disease while 3 were supported for the multisystem inflammatory syndrome in children (MIS-C) that follows primary infection. In pediatrics, MIS-C is increasingly reported; a systematic review that included 16 studies and 505 children describes 5.3% requiring ECMO [49]. These patients represent new challenges in initiation and conduct of ECMO, although recent survey data suggest most centers would offer ECMO to pediatric patients with SARS-CoV-2 related disease [13].

### 4.6. Extracorporeal Cardiopulmonary Resuscitation

The use of ECMO during acute resuscitation—extracorporeal cardiopulmonary resuscitation (ECPR)—is becoming increasingly frequent [4,50] in children, specifically for inpatient cardiac arrest [50,51]. A recent meta-analysis of pediatric studies following ECPR reported cumulative outcomes following ECPR, with odds of survival ranging between 2.5 and 3.8 for most studies except for one notable exception with odds ratio of 0.19 favoring no ECPR [52]. The largest study in this meta-analysis, comprised of 3756 patients, published by Lasa et al. reported improved survival with ECPR as compared to conventional CPR with survival rates of 40% as compared to 27% [50]. Similarly, neurologic outcomes were more favorable in the ECPR group with 27% as compared to 18% in the conventional CPR group. Interestingly, a substantial proportion of those patients who had ECPR were in the context of cardiac surgery (59% of ECPR). In contrast, Bembea et al. analyzed trends utilizing the Extracorporeal Life Support Organization and American Heart Association Get with the Guideline registries [53]. In this cohort, the rate of ECPR increased over time but survival did not, despite a similar proportion of cardiac surgical patients (59%). The overall survival between the study period of 2000 and 2014 was 41% (range 37 to 44%). A noncardiac diagnosis and preexisting renal insufficiency were associated with increased odds of death (adjusted odds ratio, 1.85 [95% CI, 1.19–2.89] and 4.74 [95% CI, 2.06–10.9], respectively).

There is little data to understand why the survival and neurological outcomes are different in the cardiac surgical patients. Potential factors include the ability to intervene if there is an anatomic or functional cardiac problem, team organization and early deployment of ECPR, or more ready recognition of cardiac decompensation. The most recent pediatric life support guideline recommendations do not support the of ECPR for out-of-hospital cardiac arrest, but suggest that it may be used for in-hospital cardiac arrest if there are conditions such as rapid access to ECLS equipment and skilled ECLS personnel as there are improved neurologic outcomes. [14]

Patients with hypoplastic left heart syndrome (HLHS) or other single ventricle physiology lesions represent a special population at risk for cardiac arrest who are difficult to resuscitate with conventional CPR. Conventional CPR may not be able to achieve flows required to support parallel circulations. A study published in 2014 reports 293 infants with HLHS who had ECPR with survival at 36% and neurologic injury noted in 16% of survivors as compared to 24% of non-survivors [54]. Characteristics of non-survivors were lower gestational age, age at cannulation, and lower body weight. Pre-cannulation physiologic parameters such as blood gas measurements, FIO_2_, and SaO_2_ did not distinguish survivors from non-survivors.

## 5. Surgical Indications for VA ECMO

### 5.1. Peri-Operative Use of VA ECMO

Survival for children with congenital heart disease has progressively improved, and one of the reasons has been widely available use of ECMO. According to the 2019 Extracorporeal life support ELSO report, HLHS was the most common congenital cardiac lesion supported by ECMO in neonates [4]. In children with congenital heart disease, those with cyanotic congenital heart lesions with decreased pulmonary blood flow (Tetralogy of Fallot, Ebstein’s anomaly of the tricuspid valve) had most frequent use of VA ECMO support [4].

### 5.2. Pre-Operative Indications

ECMO support is used in neonates pre-operatively when they present with profound cyanosis, shock, or end-organ dysfunction. In Ebstein’s anomaly and functional pulmonary atresia, VA ECMO helps to stabilize the shock state secondary to circular shunt. Similarly, in d-Transposition of great arteries, with significant pulmonary hypertension and profound cyanosis, there are reports of successful use of VA ECMO for pre-operative stabilization [55].

In children, VA ECMO is most commonly used in the acute postoperative period after congenital cardiac surgery. The frequent indications are:Failure to separate from cardiopulmonary bypass in the operating room.Low cardiac output state in the immediate post-operative period secondary to ventricular dysfunction, pulmonary hypertension or intractable arrhythmias recalcitrant to medical therapies.Cardiopulmonary arrest occurring in the acute post-operative period.

Rapid deployment of VA ECMO in patients undergoing ECPR was first reported in 1992 and has resulted in a decline in operative mortality in patients after complex cardiac surgeries [56]. Indications, outcomes and special considerations are summarized in Table 1.

### 5.3. Trends and Outcomes

Utilization of peri operative ECMO varies among centers performing congenital cardiac surgery. An analysis of the Society of Thoracic Surgeons (STS) Congenital Heart Surgery database revealed that 2.4% of patients were supported by VA ECMO after cardiac surgery from 2000 to 2017 [22]. Neonates undergoing Norwood palliation operation were more frequently supported by ECMO (17%) followed by complex biventricular repair surgeries (14%). In the same study, risk factors for post-operative ECMO included young age, low weight, mechanical ventilation prior to surgery, arrhythmia, shock, higher complexity as indicated by ‘STAT’ category of 4–5, and duration of cardiopulmonary bypass. Although patients with all forms of congenital heart disease can be rescued by ECMO, it is more frequently used in children undergoing more complex lesions or surgeries (Norwood surgery, repair of truncus arteriosus, arterial switch operation combined with VSD and arch repair) [22]. VA ECMO has been successfully used to support children of all ages and all sizes after cardiac surgery, although the smaller vasculature in premature and low birthweight babies pose extreme challenges in determining optimal cannula size and positioning. The timing of ECMO deployment and intervention for residual lesions were not investigated in this study.

Lorusso et al. performed a comprehensive review and summarized the current literature regarding use of VA ECMO after surgery for congenital heart disease [57]. They report wide variability in the use, with survival to decannulation and hospital discharge consistently between 49–58%. Single center studies report higher survival than typically found in ELSO registry reports of pediatric cardiac population [9,56]. The variability in survival varied by age, weight, complexity of the underlying cardiac surgery. Prematurity and low birthweight have been associated with increased mortality and higher incidence of neurologic complications [58].

Ford et al. analyzed the ELSO registry for outcomes of neonates supported with VA ECMO who had medical or surgical cardiac diagnoses [59]. Of the 4471 neonates, all patients weighing less than or equal to 1.5 kg and supported with ECMO died, and mortality was 75% among the patients who weighed 1.5–2 kg. Bhat et al. report survival of 33% in low-birth-weight neonates weighing ≤3 kg (median gestational age of 38 weeks) supported with VA ECMO after corrective or palliative cardiac surgery [24]. Survival for post cardiotomy patients was 41% and survival for nonsurgical patients was 42%. For patients who received ECMO following cardiac surgery, mortality was highest among those undergoing higher complexity procedures (RACHS-1 category 6) [59]. Allan et al. evaluated the indications for VA ECMO support in single ventricle patients with shunts and reported survival of 81% in those cannulated for hypoxemia secondary to shunt occlusion versus 29% in those cannulated for hypotension [25].

Mascio et al. reported that in-hospital mortality was highest after repair of truncus arteriosus and Norwood palliation for HLHS [22]. Factors that may contribute to this include right ventriculotomy, the presence of aortic suture lines that could exacerbate bleeding on VA ECMO, and the potential for coronary insufficiency. Many studies have identified other predictors for in-hospital mortality, which are collectively summarized in Table 3 [15,26,27,28,29,30,31,32,60].

### 5.4. Residual Lesions

Multiple studies have emphasized the importance of early cardiac catheterization aimed at identifying and treating residual lesions to facilitate timely liberation from ECMO. Agarwal et al. reported the presence of residual lesions in approximately 25% of post-operative cardiac surgery patients receiving VA ECMO support, thus concluding that those unable to be weaned off VA ECMO should have detailed evaluation for residual lesions, preferably by cardiac catheterization in addition to echocardiography [61]. Multiple studies confirm that early cardiac catheterization, identification and/or correction of residual lesion is associated with shorter duration of ECMO and survival [19,63]. Cardiac catheterization can be safely performed on patients supported by ECMO, and is a critical tool in the early recognition, diagnosis, and treatment of hemodynamically significant anatomic abnormalities.

### 5.5. Surgical Indications for VA ECMO in the Single Ventricle Pathway

#### Stage 1 Palliation

Children living with functionally single ventricle physiology undergo stage 1 palliative surgery involving the construction of neo-aorta and creation of pulmonary blood flow with a systemic to pulmonary artery shunt and atrial septectomy. Following their stage 1 surgery for HLHS, 13–20% have been supported by ECMO [20,21,64]. In addition to low birthweight, certain specific risk factors that predicted the need for ECMO support were identified as long cardiopulmonary bypass time, ascending aorta < 2 mm, the subset of mitral stenosis/aortic atresia patients, and intraoperative shunt revision. In the Single Ventricle Reconstruction trial, after adjusting for surgeon and birth weight, neonates with a modified Blalock Taussig shunts had smaller chances of receiving ECMO and had significantly better outcome after ECPR or ECMO compared to neonates with Right Ventricle-Pulmonary artery shunts [21]. Allan et al. reported in a single-center study in 44 neonates, patients cannulated for hypoxemia, and particularly shunt obstruction, had markedly improved survival (83%) compared to hypotension (29%) [59]. In the single-ventricle reconstruction (SVR) trial, most neonates (70%) received ECMO after stage 1 because they could not be separated from cardiopulmonary bypass [64]. Sherwin et al. reported survival to discharge in HLHS after Norwood surgery supported by ECMO as 31% [65]. Predictors of mortality identified were pre-ECMO ventilation > 5 days, pre-ECMO PEEP > 8, and increased ECMO duration. In a single-center study by Hoskote et al., 56% of post-operative single-ventricle ECMO patients received ECMO because of cardiac arrest, and 44% because of low cardiac output syndrome, with 44% survival to discharge [66].

### 5.6. Bidirectional Superior Cavopulmonary Anastomoses—Glenn Surgery

While the outcomes for the Glenn procedure are excellent, patients are still at risk of decompensation and may need ECMO support. The Glenn surgery results in separation of superior vena caval pathway from the heart, so the venous drainage site must be carefully considered. Some centers routinely cannulate superior vena cava (SVC) first to preserve cerebral perfusion and after initial stabilization, decide on additional venous drainage cannulas or transition to central cannulation.

Booth et al. described a single center series of 6 patients supported by VA ECMO post Glenn surgery with only 3 successfully decannulated who suffered significant neurological injury [67]. In the study by Mascio et al. on the use of VA ECMO in single ventricle patients, Glenn physiology represented only 0.8% [55]. Jolley et al. described VA ECMO use in 103 patients with Glenn physiology [6]. Overall survival was 41% with neurological sequelae such as seizure, hemorrhagic stroke or embolic stroke in 23% of total patients and 14% of survivors. Predictors of mortality were identified as presence of renal failure neurological injury and persistent acidosis. Although overall mortality is reportedly lower compared to the previous data, this subset of patients continues to have significant morbidity while supported on VA ECMO [6].

### 5.7. Total Cavopulmonary Anastomoses—Fontan Surgery

After Fontan operation, the inferior vena cava (IVC) is incorporated into the superior cavopulmonary anastomosis, so the passive pulmonary blood flow is followed by the pulmonary venous return to the systemic ventricle. This physiology can fail either acutely or gradually with impaired hemodynamics and end organ perfusion necessitating ECMO support. Critical to support of the patient with Fontan physiology is decompression of the Fontan circuit with maintenance of adequate perfusion pressure. To accomplish these goals, additional venous cannula (upper and lower) or central cannulation may be necessary. Additionally, venous stasis in the Fontan circuit should be avoided. To address this, some centers consider a veno-arterial-venous strategy, in which there is there is one drainage cannula and two return cannulae. One of the return cannula is placed in the carotid and the other in the internal jugular. This allows for flow of oxygenated blood into the caval system with simultaneous decompression preventing stasis.

Rood et al. describes the largest cohort of 230 patients in Fontan physiology who were supported by ECMO [68]. 56% of the cohort were able to be liberated from ECMO but only 35% survived to hospital discharge. Five patients in that series required heart transplant. Another series from single center reported survival of 50% in the Fontan cohort [67]. The survival was affected by complications of ECMO such as bleeding, neurological complications and renal failure.

Fontan failure can be classified into early, mid and late phases. Early failure is often due to cavopulmonary obstruction that may be relieved surgically with short term rescue using VA ECMO. Mid and late phase failure often present with end-organ dysfunction, protein losing enteropathy, or plastic bronchitis, that present significant risk factors for VA ECMO and carry even higher risk of mortality. Although most of the focus of the mid and late phase Fontan failure patients has shifted towards VAD, the outcomes continue to remain grim in this cohort.

### 5.8. Hybrid Palliation

Hybrid procedure is palliative and combines bilateral pulmonary artery bands and stent to maintain patency of ductus arteriosus for patients who are considered to have higher risk of mortality with Norwood operation. Roeleveld et al. identified 44 patients with HLHS and hybrid palliation who needed ECMO support [68]. The indication for VA ECMO was low cardiac output in 83% of patients, 18% had ECPR, and 50% were liberated from VA ECMO, but only 16% of them survived to hospital discharge.

A review from Mitchell et al. from a single center that selectively performs hybrid procedure as its stage 1 palliation, included 181 patients [69]. Incidence of VA ECMO after hybrid palliation was reported at 1.1%. Both the patients who were placed on VA ECMO died. In comparison, the Children’s Hospital in Giessen published their 15-year experience with hybrid palliation, which is their standard approach to all patients with single ventricle patients [69]. Mitchell et al. concluded in their study that in an institution that only performs hybrid procedure for stage 1 single ventricle physiology, mortality in patients who were placed on VA ECMO after hybrid procedure was higher than reported for Norwood procedure, while the incidence of ECMO was also lower than reported for Norwood operation. Future studies are warranted to understand if VA ECMO is a reasonable option for this cohort.

### 5.9. Surgical Indications for VA ECMO after Heart Transplantation

Heart transplant is a well-established treatment for heart failure refractory to medical therapies. Acute graft failure represents the most common reason for VA ECMO post-operatively and may occur for many reasons [16,17,70]. When medical therapies to target each etiology fail, VA ECMO or VAD bridge to recovery or re-transplantation. Tissot et al. reported 9% of heart transplants utilizing ECMO for primary graft failure [71]. 54% were decannulated and alive to hospital discharge, 100% survival at 3 years, and 46% survival at 8-year follow-up. The patients who went on VA ECMO were younger, weighed less, and had longer ischemic times compared to patients who did not need VA ECMO and decreasing survival with duration of cannulation. A similar study by Godown et al. using a linked PHIS/SRTR database of pediatric heart transplant recipients reported that 7.9% required VA ECMO after transplant, of whom 85.3% were decannulated and 87.4% survived to hospital discharge [72]. The risk factors identified in those requiring VA ECMO support included younger age, congenital heart disease, VA ECMO prior to transplant, mechanical ventilation prior 74to transplant, decreased estimated creatinine clearance, and ischemic time greater than 4 h. The long-term survival in these patients was no different compared to patients who did not require ECMO support. These data continue to support the use of ECMO in the early post-transplant period.

### 5.10. Special Considerations

While ECMO effectively restores systemic organ perfusion to end organs in cardiogenic shock, it has detrimental effects on the left ventricle. When the aortic valve fails to open, the ongoing venous return, via thebesian and bronchial veins, leads to LV distension. This cascades to impaired myocardial perfusion pressure, intra-cardiac thrombus, dysrhythmias, and LA hypertension with resultant pulmonary hemorrhage, and hinders myocardial recovery. Measures to facilitate LV ejection with inotropic support and decreasing mean arterial blood pressure may help, but decompressing the left heart with an atrial septostomy or LA vent may be required.

There are no defined diagnostic criteria for LV distension, although lack of pulsatility via arterial line, clinical and radiographic evidence of significant pulmonary edema, elevated pulmonary capillary wedge pressure, echocardiographic evidence of LV distension and stasis, intermittent or absent opening of the aortic valve aid in detecting the LV distension. There are currently no consensus guidelines to delineate the optimal timing and method of decompression, although the decompression has been accomplished by surgical and percutaneous methods.

Xie et al. performed a metanalysis investigating the current evidence for incidence, diagnosis, prevention, and intervention for LV overload in both adult and pediatric patients [62]. The incidence is highly variable, ranging from 1–68%, although this is a combined data of adult and pediatric patients. Some centers perform this prophylactically, especially in the pediatric population as infants have lower myocardial compliance, making them vulnerable to LV distension sequelae [73,74]. The techniques available for LV decompression can be summarized in Table 4 [62].

Zampi et al., in a retrospective multicenter study of pediatric patients receiving VA ECMO, observed that late left atrial decompression (≥18 h) was associated with longer duration of ECMO support and mechanical ventilation [75]. Early decompression compared to late did not have survival benefit, but morbidities of long VA ECMO run may justify atrial decompression.

## 6. Conclusions

We have described the major surgical and medical indications for use of VA ECMO for cardiogenic shock. The overall use of VA ECMO has increased over time and the number of indications has expanded. Despite the growing familiarity with VA ECMO and technical advances, survival and other outcome measures have not improved. The potential complications of VA ECMO limit the use of VA ECMO to weeks at best. When quick recovery is expected, the risk factors associated with VA ECMO do not impose limitations, but when recovery is longer or requires bridge to diagnosis or transplant, these factors are more relevant. Transition to VAD may prove to be a better strategy [43]. Several studies report that patients supported with VAD compared to ECMO have longer, superior survival and superior post-transplant outcomes [76,77]. Some of the critical decisions that need to be made include the intent of the device support and timing of transition from VA ECMO to VAD such that the patient is safely bridged to recovery or transplant. Byrnes at al. have described an algorithm that may help with this decision-making process at different time points [78].

## 7. Future Directions

How can we optimize outcomes in VA ECMO? Perhaps we should have a more restrictive thought process related to candidacy. In our institution (Children’s Medical Center at Dallas), candidacy for VA ECMO and ECPR are considered independently for all patients in the cardiovascular intensive care unit and discussed in a multidisciplinary format. Literature provides some guidance on which patients will have a positive outcome, but other considerations include not just survival, but developmental sequelae, or parental perceptions of all forms of extracorporeal support. More research is needed to understand the long-term sequela of VA ECMO. In those patients for whom we offer VA ECMO or ECPR, timing is critical and should avoid end-organ dysfunction in which there is no hope of recovery. SARS-CoV-2 associated disease remains a novel front for VA ECMO and requires significant study. The single ventricle population, especially after Glenn and Fontan surgery, remains at high risk of mortality and morbidity with ECPR, and their candidacy for ECPR has to be carefully evaluated.

## Figures and Tables

**Figure 1 jcm-10-01573-f001:**
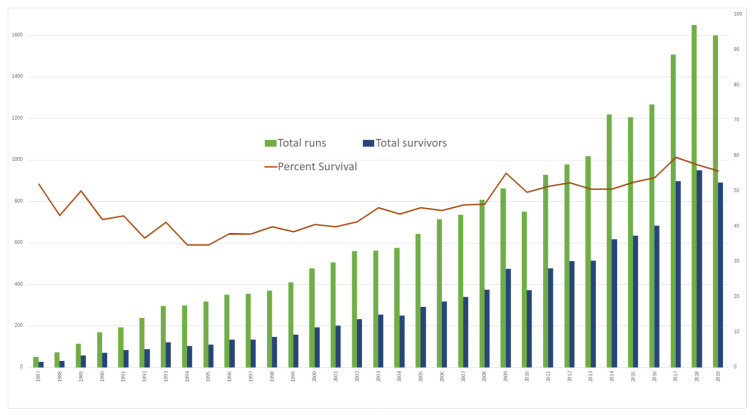
Number of neonatal and pediatric VA ECMO runs per year with percent survival. Adapted from the ELSO Registry International Summary [4].

**Table 1 jcm-10-01573-t001:** Summary of Medical and Surgical Indications and Outcomes for VA ECMO.

Indications		Survival to Hospital Discharge after ECMO	Notes
Medical	Myocarditis	21.4–83.3% survival to hospital discharge [6,7]	Incidence 36% [6]
	Cardiomyopathy	61% survival to hospital discharge [5]	Incidence 28% in patients with cardiomyopathy and listed for transplant [8]
	Intractable arrhythmia	100% survival to hospital discharge in single center study (*n* = 9) [9]	Often associated with cardiomyopathies or myocarditis
	Sepsis	59% survival to hospital discharge [10]	Central cannulation may be necessary to maximize support
	Novel reasons	50% survival to hospital discharge (*n* = 6) [11]	SARS-CoV-2 is a novel indication for both primary respiratory disease and MIS-C [11]
	Cardiopulmonary arrest	37–41% survival to hospital discharge [12,13]	59% of ECPR was after cardiac surgery [13]
Surgical	Peri-operative	49–58% survival to hospital discharge [14]; Failure to separate from cardio-pulmonary bypass: survival 45% to hospital discharge [15]	Incidence 2.4% [14]
	Single ventricle palliation	Please see Table 2	
	Heart transplantation	54–87.4% survival to hospital discharge [16,17]	Overall incidence 7.9–9%; Primary graft failure is the most common indication [16,17]

**Table 2 jcm-10-01573-t002:** Summary of VA ECMO Strategies and Survival in Stages of Single Ventricle Palliation

Single Ventricle Palliation Stage	Cannulation Strategy	Survival to Hospital Discharge	Notes
Hybrid	Internal jugular Carotid	ECMO cannulation in 2/149 patients with hybrid procedure, both died [18]	Early conversion to central or proceed to single ventricle palliation if possible
Norwood	Central	Incidence of ECMO 13–20%; survival 29–81% with highest survival in BT * shunt occlusion [19,20,21]	May require high flows for parallel circulation
Bidirectional Glenn	Internal jugular Carotid	Incidence 0.8% [22]; survival 41% [6]	Early conversion to central
Fontan	Internal jugular Femoral vein Carotid	Incidence 4% [23] survival 35%	Early conversion to central; alternative cannulation strategy includes veno-arterial-venous strategy

* Blalock-Taussig.

**Table 3 jcm-10-01573-t003:** Predictors of mortality in VA ECMO for Surgical Indications

Predictors of Mortality
Prematurity & younger age [24,27,28,55,59]
Low bodyweight (<3 kg) [26,54,55]
Single-ventricle physiology [26,54,59,61]
Pre-ECMO high inotrope score [26]
Pre-ECMO mechanical ventilation [27,54]
Pre ECMO acidosis [25,54,55]
Failure to clear lactate on ECMO [25,29,62]
Renal failure & fluid overload on ECMO initiation [25,27,29]
Bleeding during ECMO and need for PRBC transfusion [25]
Duration of ECMO support > 7 days [25,30,55]

**Table 4 jcm-10-01573-t004:** Summary of left ventricular decompression techniques. Adapted from Xie et al. [62].

Summary of Left Ventricular Decompression Techniques
Surgical
Right upper pulmonary vein, with tip terminating in LA or LV
Pulmonary artery
LA appendage
LV apex
Percutaneous
Transseptal venting
Transseptal needle puncture
Balloon septostomy
Blade septostomy
Transseptal cannulation
Transpulmonary venting
Ventricular assist devices
Impella
Intra-aortic balloon pump

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
