# Peer review of "Use of ECMO for Cardiogenic Shock in Pediatric Population"

_jcm, 2021, doi:10.3390/jcm10081573_

Round 1

Reviewer 1 Report

In overall, the manuscript is lengthy and too wordy. It could be considered to be truncated by 20-30% without losing the contents. Focus on “ECMO for pediatric cardiogenic shock”.

Making tables showing literatures with information like subjects, indications, outcomes or specific notes for each indication categories could be considered. It would help reducing words in the text.

Delete notions for PPHN, as it is out of the scope for “ECMO for pediatric cardiogenic shock”.

Providing some figures for the schema of vascular access and circulation for each congenital anomalies; i.e. HLHS before stage I, after Norwood, after Glenn, after Fontan or after Hybrid. It would help better understanding for the readers.

L434-438 is hard to understand. Add any explanations (or a figure if needed).

Did the authors obtain copyright permission for the figure 1?

Figure 2 could be expressed as a table.

Figure 3 would be unnecessary. No citation in the text.

Conclusions are too long. Please concisely summarize the contents of the text.

Many typos could be found.

Reviewer 2 Report

The manuscript entitled ‘Use of ECMO for cardiogenic shock in pediatric population’ by Bhaskar et al provides a comprehensive review for ECMO use in pediatric population. However, there several issues to be addressed.

  • the abstract the authors claim that This review provides an overview of 12 VA ECMO used f after surgery for congenital heart disease. However, this is not correct since the manuscript provides a more wide spectrum in this issue and the sentence should be rephrased.
  • The rate of of ECMO use in myocarditis at 36% might be misleading since there other studies estimated this rate much lower. Authors should refer the range that is reported in the literature
  • A short discussion should be added regarding the predictors for ECMO use and ECMO outcomes.
  • Types of ECMO should be given and explain their indications on each case.
  • A brief description of EXCOR device should be given when it is first presented in the text.
  • A focus on areas for future development should be given
  • Hemostatic complications and management should be discussed
  • An emphasis oh the guidelines for pediatric life support should be aded

Minor comments

  • [1] Provide the abbreviation for ECLS
  • [2] Since all the indications of ECMO are medical, this square should be changed in `figure 2.

Round 2

Reviewer 2 Report

Although I received the reply letter for the other reviewer I think that the authors have responded adequately to the my comments